# Nonregistration, discontinuation, and nonpublication of randomized trials: A repeated metaresearch analysis

Benjamin Speich[1,2‡*], Dmitry Gryaznov[1‡], Jason W. Busse[3,4], Viktoria L. Gloy[1], Szimonetta Lohner[5,6], Katharina Klatte[7], Ala Taji Heravi[1,8], Nilabh Ghosh[1], Hopin Lee[2], Anita Mansouri[2], Ioana R. Marian[2], Ramon Saccilotto[7], Edris Nury[9,10], Benjamin Kasenda[11], Elena Ojeda–Ruiz[1,12], Stefan Schandelmaier[1,3], Yuki Tomonaga[13], Alain Amstutz[1,8,14], Christiane Pauli–Magnus[7], Karin Bischoff[15,16], Katharina Wollmann[16], Laura Rehner[15,17], Joerg J. Meerpohl[15,16], Alain Nordmann[1], Jacqueline Wong[3], Ngai Chow[3], Patrick Jiho Hong[3,18], Kimberly Mc Cord – De Iaco[1,19], Sirintip Sricharoenchai[1], Arnav Agarwal[3,20], Matthias Schwenkglenks[13,21], Lars G. Hemkens[1,22,23], Erik von Elm[24], Bethan Copsey[2], Alexandra N. Griessbach[1], Christof Schönenberger[1], Dominik Mertz[3,25], Anette Blümle[8,26], Belinda von Niederhäusern[6,27], Sally Hopewell[2], Ayodele Odutayo[2,28], Matthias Briel[1,3]

1 Meta–Research Centre, Department of Clinical Research, University Hospital Basel, University of Basel, Basel, Switzerland, 2 Oxford Clinical Trials Research Unit / Centre for Statistics in Medicine, Nuffield Department of Orthopaedics, Rheumatology and Musculoskeletal Sciences, University of Oxford, Oxford, United Kingdom, 3 Department of Health Research Methods, Evidence, and Impact, McMaster University, Hamilton, Canada, 4 Department of Anesthesia, McMaster University, Hamilton, Canada, 5 Cochrane Hungary, Clinical Centre of the University of Pécs, Medical School, University of Pécs, Pécs, Hungary, 6 Department of Public Health Medicine, Medical School, University of Pécs, Pécs, Hungary, 7 Clinical Trial Unit, Department of Clinical Research, University of Basel and University Hospital Basel, Basel, Switzerland, 8 Swiss Tropical and Public Health Institute, Basel, Switzerland, 9 Institute for Evidence in Medicine (for Cochrane Germany Foundation), Faculty of Medicine and Medical Center, University of Freiburg, Freiburg, Germany, 10 Department of General Practice and Primary Care, Medical Center Hamburg–Eppendorf–UKE, Hamburg, Germany, 11 Department of Medical Oncology, University of Basel and University Hospital Basel, Basel, Switzerland, 12 Bioaraba Health Research Institute, Health Prevention, Promotion and Care Area; Osakidetza Basque Health Service, Araba University Hospital, Preventive Medicine Department, Vitoria–Gasteiz, Spain, 13 Epidemiology, Biostatistics and Prevention Institute, University of Zurich, Zurich, Switzerland, 14 Department of Infectious Diseases and Hospital Epidemiology, University Hospital Basel, Basel, Switzerland, 15 Institute for Evidence in Medicine, Medical Center–University of Freiburg, Faculty of Medicine, University of Freiburg, Freiburg, Germany, 16 Cochrane Germany, Cochrane Germany Foundation, Freiburg, Germany, 17 Institute for Nursing Science and Interprofessional Learning, University Medicine Greifswald, Greifswald, Germany, 18 Department of Anesthesiology and Pain Medicine, University of Toronto, Toronto, Canada, 19 Multifactorial and Complex Diseases Research Area, Bambino Gesù Children's Hospital IRCCS, Rome, Italy, 20 Division of General Internal Medicine, Department of Medicine, McMaster University, Hamilton, Ontario, Canada, 21 Institute of Pharmaceutical Medicine (ECPM), University of Basel, Basel, Switzerland, 22 Meta–Research Innovation Center Berlin (METRICS–B), Berlin Institute of Health, Berlin, Germany, 23 Meta–Research Innovation Center at Stanford (METRICS), Stanford University, Stanford, California, United States of America, 24 Cochrane Switzerland, Centre for Primary Care and Public Health (Unisanté), University of Lausanne, Lausanne, Switzerland, 25 Department of Medicine, McMaster University, Hamilton, Canada, 26 Clinical Trials Unit, Faculty of Medicine and Medical Center, University of Freiburg, Freiburg, Germany, 27 Roche Pharma AG, Grenzach–Wyhlen, Germany, 28 Applied Health Research Centre, Li Ka Shing Knowledge Institute of St Michael's Hospital, Toronto, Ontario, Canada

‡ These authors share first authorship on this work.
* benjamin.speich@usb.ch

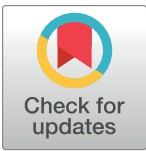

**Data Availability Statement:** All participating ethics committees were project partners and

granted us access to confidential study protocols under the condition that only aggregated data will be made publicly available. All investigators signed a corresponding confidentiality agreement with ethics committees. All aggregated data that were used for this study are presented in Tables and Figures of the manuscript or the appendix. The appendix also includes the used statistical code (STATA; S5 Text).

**Funding:** The study is supported by the Swiss Federal Office of Public Health(MB). BS is supported by an Advanced Postdoc.Mobility (P300PB_177933), a Return Postdoc.Mobility (P4P4PM_194496) grant from the Swiss National Science Foundation, and a personal grant from the Research Foundation of the University of Basel. IRM holds a NIHR Pre-Doctoral Fellowship. SL was supported by the Alexander von Humboldt Foundation, Germany during her research stay at the Institute for Evidence in Medicine, University of Freiburg, Germany and is currently supported by the János Bolyai Research Scholarship of the Hungarian Academy of Sciences (BO/00498/17/5). The funders had no role in study design, data collection and analysis, decision to publish, or preparation of the manuscript.

**Competing interests:** I have read the journal's policy and the authors of this manuscript have the following competing interests: DG contributed to the ASPIRE project as part of his PhD thesis before his current employment with Idorsia Pharmaceuticals Ltd. (his current employer had no role in study design, data collection and analysis, decision to publish, or preparation of the manuscript). BvN contributed to the ASPIRE project as part of her PhD thesis before her current employment with Roche (her current employer had no role in study design, data collection and analysis, decision to publish, or preparation of the manuscript). All authors have declared that no competing interests exist.

**Abbreviations:** ASPIRE, Adherence to SPIrit REcommendations; CI, confidence interval; CRO, Contract Research Organization; CTU, Clinical Trial Unit; IQR, interquartile range; OR, odds ratio; PRISMA, Preferred Reporting Items for Systematic Reviews and Meta-Analyses; RCT, randomized clinical trial; REC, research ethics committee; SPIRIT, Standard Protocol Items: Recommendations for Interventional Trials.

# Abstract

## Background

We previously found that 25% of 1,017 randomized clinical trials (RCTs) approved between 2000 and 2003 were discontinued prematurely, and 44% remained unpublished at a median of 12 years follow-up. We aimed to assess a decade later (1) whether rates of completion and publication have increased; (2) the extent to which nonpublished RCTs can be identified in trial registries; and (3) the association between reporting quality of protocols and premature discontinuation or nonpublication of RCTs.

## Methods and findings

We included 326 RCT protocols approved in 2012 by research ethics committees in Switzerland, the United Kingdom, Germany, and Canada in this metaresearch study. Pilot, feasibility, and phase 1 studies were excluded. We extracted trial characteristics from each study protocol and systematically searched for corresponding trial registration (if not reported in the protocol) and full text publications until February 2022. For trial registrations, we searched the (i) World Health Organization: International Clinical Trial Registry Platform (ICTRP); (ii) US National Library of Medicine (ClinicalTrials.gov); (iii) European Union Drug Regulating Authorities Clinical Trials Database (EUCTR); (iv) ISRCTN registry; and (v) Google. For full text publications, we searched PubMed, Google Scholar, and Scopus. We recorded whether RCTs were registered, discontinued (including reason for discontinuation), and published. The reporting quality of RCT protocols was assessed with the 33-item SPIRIT checklist. We used multivariable logistic regression to examine the association between the independent variables protocol reporting quality, planned sample size, type of control (placebo versus other), reporting of any recruitment projection, single-center versus multicenter trials, and industry versus investigator sponsoring, with the 2 dependent variables: (1) publication of RCT results; and (2) trial discontinuation due to poor recruitment.

Of the 326 included trials, 19 (6%) were unregistered. Ninety-eight trials (30%) were discontinued prematurely, most often due to poor recruitment (37%; 36/98). One in 5 trials (21%; 70/326) remained unpublished at 10 years follow-up, and 21% of unpublished trials (15/70) were unregistered. Twenty-three of 147 investigator-sponsored trials (16%) reported their results in a trial registry in contrast to 150 of 179 industry-sponsored trials (84%).

The median proportion of reported SPIRIT items in included RCT protocols was 69% (interquartile range 61% to 77%). We found no variables associated with trial discontinuation; however, lower reporting quality of trial protocols was associated with nonpublication (odds ratio, 0.71 for each 10% increment in the proportion of SPIRIT items met; 95% confidence interval, 0.55 to 0.92; $p = 0.009$). Study limitations include that the moderate sample size may have limited the ability of our regression models to identify significant associations.

## Conclusions

We have observed that rates of premature trial discontinuation have not changed in the past decade. Nonpublication of RCTs has declined but remains common; 21% of unpublished trials could not be identified in registries. Only 16% of investigator-sponsored trials reported results in a trial registry. Higher reporting quality of RCT protocols was associated with

publication of results. Further efforts from all stakeholders are needed to improve efficiency and transparency of clinical research.

## Author summary

### Why was this study done?

- Registration of a clinical trial is crucial to prespecify outcomes, deter publication bias, and avoid unnecessary duplication of clinical research.

- The rate of prematurely discontinued trials (primarily due to preventable reasons) and nonpublished trials was high in a study conducted 10 years ago.

- An assessment, providing robust data on the proportion of randomized trials that are registered, and an update to explore if the proportion of discontinued and nonpublished trials has changed, was warranted.

### What did the researchers do and find?

- We assessed whether 326 study protocols of randomized clinical trials that received ethical approval in 2012 (in Switzerland, the United Kingdom, Germany, and Canada) were registered, completed, and published.

- We found that 6% (19/326) of trials were unregistered, 30% (90/326) were prematurely discontinued with poor recruitment as the most common reason, and 21% (70/326) were not published.

- While only 2% (254/256) of published randomized trials were not registered, among nonpublished trials 21% (55/70) were not registered.

- While results are available on a trial registry for 84% (150/179) of industry-sponsored trials, this is only the case for 16% (23/147) of investigator-sponsored trials.

### What do these findings mean?

- This metaresearch study revealed that discontinuation of randomized trials is still common, contributing considerably to research waste as the most common reasons for discontinuation appear preventable.

- In terms of making trial results available, the situation has improved during the last 10 years.

- Trialists should be encouraged to demonstrate feasibility before embarking on definitive trials, and further efforts are required to improve reporting of trial results in registries for investigator-sponsored trials.

## Introduction

Rigorously planned and conducted randomized clinical trials (RCTs) are critical to inform the effectiveness and safety of healthcare interventions [1,2]. Clinical trial registries were implemented in the early 2000s to avoid unnecessary duplication of research and to estimate and deter publication bias. In 2005, the International Committee of Journal Editors proclaimed prospective trial registration (i.e., registration before enrolling the first participant) as a requirement for publication [3,4]. Shortly thereafter, laws in different regions (e.g., European Union and North America) required trials to be registered [5,6] and making results available [7,8]. Several studies have explored the proportion of published RCTs that are registered [9–12]; however, such investigations used published RCTs—and not RCT protocols—as the denominator and thus were unable to assess publication bias.

Trial discontinuation and nonpublication can constitute substantial research waste [13]. For example, if an RCT is discontinued due to slow participant recruitment before the planned sample size is reached, the trial is typically not sufficiently powered to answer the primary research question. The data, however, can still be useful in meta-analyses. Hence, it is crucial that all RCT results, including discontinued trials, are made available so that evidence is not lost and resource waste is minimized. We conducted an international metaresearch study of 1,017 RCT protocols approved between 2000 and 2003 that found 1 in 4 trials was prematurely discontinued, primarily due to poor recruitment [14]. Only 59% of approved trials were published at a median follow-up of 12 years, and premature discontinuation was associated with a lower likelihood of publication [14]. We acquired a new sample of RCT protocols, approved in 2012 by the same research ethics committees (RECs) plus an REC from the United Kingdom, to explore if trial completion and publication rates have changed. In addition, we aimed to investigate to what extent nonpublished RCTs can be found in trial registries, and the association between reporting quality of RCT protocols and trial discontinuation due to poor recruitment or nonpublication of results [15].

## Methods

The present study is an associated project of the Adherence to SPIrit REcommendations (ASPIRE) study [15]. The ASPIRE study group is an international collaboration of researchers with a mandate to evaluate the completeness of RCT protocols before and after publication of the Standard Protocol Items: Recommendations for Interventional Trials (SPIRIT) statement [16]. Further ASPIRE substudies examine the use of patient-reported outcomes in RCT protocols, the reporting quality of RCT protocols with regulated versus nonregulated interventions, the planning of subgroup analyses in RCT protocols, and the use of routinely collected data in RCTs [15].

### Study sample

The rationale and protocol for this study has been published [15]. In brief, we acquired 360 RCT protocols that were approved in 2012 by RECs located in Switzerland (Basel, Bellinzona, Bern, Geneva, Lausanne, St. Gallen, and Thurgau), the UK (the Bristol office of the UK National Research Ethics Service responsible for 19 RECs in the UK), Germany (Freiburg), and Canada (Hamilton; see **S1 Text** for details of participating RECs). We included RCTs in which participants were randomly assigned to different interventions (or an intervention and control group) to evaluate effects on health outcomes. We included all eligible RCT protocols that were available at participating RECs with the exception of Freiburg (Germany), Hamilton (Canada), and Zürich (Switzerland), where a random sample was selected (see study protocol

for more details [15]). Studies labeled as pilot, feasibility, or phase 1 studies were excluded [15]. We also excluded duplicate protocols, and protocols for trials that were ongoing, had not started at the time of data collection, or were terminated but did not recruit any patients.

Reviewers determined, independently and in duplicate (for over 75% of included protocols), the reporting quality of all eligible RCT protocols by assessing the proportion of SPIRIT checklist items met [15–17].

## Data collection

Reviewers determined whether each RCT was registered, prematurely discontinued (including reasons for discontinuation), and if trial results were published in a peer-reviewed journal or trial registry. In detail, we assessed if trials were registered by reviewing REC files and through a systematic search of the following registries and databases between March and September 2019: (1) the World Health Organization: International Clinical Trial Registry Platform (ICTRP) database; (2) US National Library of Medicine (ClinicalTrials.gov); (3) European Union Clinical Trial Registry (EUCTR); and (4) the ISRCTN registry. We also used the Google search engine to identify registration details. We classified an RCT as unregistered if we were not able to find any record or registration through our search. We used the following strategies to identify trials: (i) searching trial registration numbers (if reported in the protocol); (ii) full titles; (iii) short titles; (iii) study acronyms; and (iv) searching for the study population and intervention (with or without specifying the control group).

We extracted the trial status (i.e., completed, early discontinuation and why, or unclear), planned and achieved sample size, availability of study results, and reported links to full text publications. In February 2022, we rechecked the trial status and availability of study results for all included RCTs. We designated the trial status as unclear when an RCT was labeled as ongoing in the registry but the status had not been updated in the previous 2 years, unless the planned completion date was after February 2022. We contacted RECs and surveyed principal investigators for clarification when trial status was unclear (see **S2** and **S3 Texts** for details). We classified RCTs as prematurely discontinued if they were specified as such in a trial registry, publication, or communication with a REC or trial author, or if the achieved sample size was <90% of the prespecified target sample size in the approved study protocol [14]. We conducted a systematic search of the following 3 electronic databases for full text publications corresponding to RCT protocols: PubMed, Google Scholar, and Scopus (see **S4 Text** for search strategies). For all corresponding full text publications identified, we extracted the planned and achieved sample size and, if applicable, the reason for premature trial discontinuation. All searches and data extraction were conducted in duplicate, and disagreements were resolved by discussion.

## Analysis

Trial registration, publication, completion, and reasons for discontinuation are reported as frequencies and percentages with 95% confidence intervals (CIs), stratified by sponsorship (industry versus investigator) and country of study approval. We conducted univariable and multivariable logistic regression analyses with the following factors as dependent variables: (i) publication in a peer-reviewed journal; (ii) premature trial discontinuation due to poor recruitment; and (iii) discontinuation due to preventable reasons (considering discontinuation due to futility, benefit, harm, and external evidence as not preventable [18]; not prespecified in protocol paper [15]). These variables were selected because all RCTs should be published and discontinuation due to recruitment problems or preventable reasons should ideally be avoided (while other reasons for discontinuation, e.g., due to benefit or harm, might be in the interest

of patients). We examined the following 7 independent variables in our models, hypothesizing that they might be associated with lower rates of discontinuation and nonpublication: (1) greater proportion of SPIRIT items reported in the protocol; (2) larger target sample size; (3) use of an active comparator versus placebo; (4) multicenter versus single-center study; (5) reporting of any recruitment projection versus not reporting; (6) industry- versus investigator-sponsored trials; and (7) support from a Clinical Trial Unit (CTU) or a Contract Research Organization (CRO) versus no support. The first variable was prespecified in our study protocol [15]. The other 6 variables were selected post hoc following our previous conducted analysis of trial protocols approved between 2000 and 2003 to facilitate comparison [14]. Variable 7 could only be included in a separate analysis in which we excluded UK protocols, as we were not able to extract from these protocols whether support from a CTU or CRO was provided. For all regression models, we calculated unadjusted and adjusted odds ratios (ORs) with 95% CIs. All analyses were conducted using Stata version 16.1 with $p < 0.05$ (2-sided) as the threshold for statistical significance (see **S5 Text** for statistical code).

### Registration and protocol

Since PROSPERO does not allow the registration of systematic metaresearch studies that do not assess the effect of an intervention on a health outcome, we did not register this study. The protocol is published and publicly available [15].

### Patient and public involvement and reporting

Patients or the public were not involved in the design, or conduct, or reporting, or dissemination plans of our research. This study is reported as per the Preferred Reporting Items for Systematic Reviews and Meta-Analyses (PRISMA) guideline [19,20].

### Results

Of the 360 potentially eligible RCT protocols approved by our study RECs in 2012, 14 were ongoing, 15 never started, and 5 were duplicate submissions, resulting in a total of 326 protocols for analysis (**Fig 1**). Included RCT protocols had a median planned sample size of 250 participants (interquartile range [IQR] 100 to 600) and a median proportion of 69% SPIRIT items reported (IQR 61% to 77%). Approximately half (55%) were industry sponsored (179/326), the majority were multicenter studies (82%; 266/326), employed a parallel group study design (91%; 296/326), and assessed the effect of a drug (64%; 207/326; **Table 1**). Most included RCT protocols were approved in Switzerland (51%; 165/326), 27% in the UK (89/326), 11% in Germany (37/326) and Canada (35/326). Baseline characteristics stratified by countries are presented in **S1 Table**. Characteristics of included trials were similar compared to our previous study of RCT protocols approved between 2000 and 2003 [14].

Of 326 RCTs, 94% (307/326) were registered (84% prospectively; 274/326, 10% retrospectively; 33/326), and 6% (19/326) of trials were unregistered (for 11 of the 19 unregistered RCTs completion status remained unclear; Table 2). Retrospective registration, meaning registration after recruitment of first patient, was more common in investigator-sponsored trials (15%; 22/147) than industry-sponsored trials (6%; 10/179). Approximately half (53%; 173/326) of all RCTs reported their results in a trial registry (industry-sponsored 84% [150/179] vs. 16% [23/147] investigator-sponsored), and 79% (256/326) of trials were published in a peer-reviewed journal at 10 years follow-up. Of the 70 RCTs that were not published in a peer-reviewed journal, 15 (21%) were unregistered. These 15 RCTs had a median planned sample size of 80 (IQR, 30–150) and were mostly single-center (67%; 10/15) investigator-sponsored (80%; 12/15) trials. Among the 256 published trials, only 4 were unregistered (2%). Results for 42 RCTs (13%)

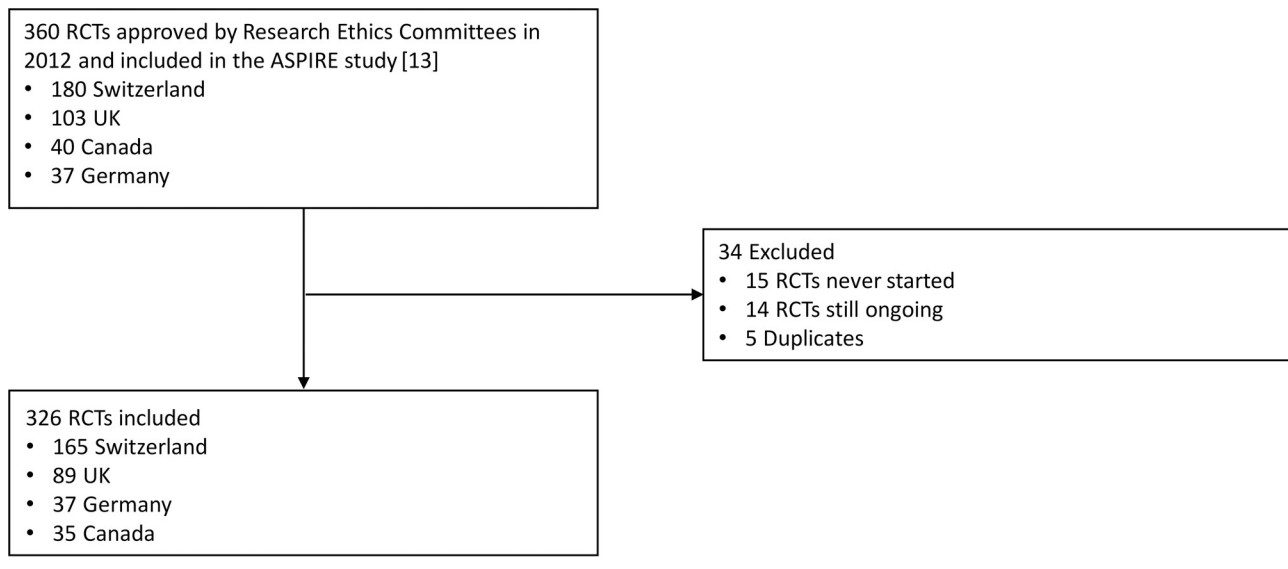

**Fig 1. Flow chart.** ASPIRE, Adherence to SPIrit REcommendations; RCT, randomized clinical trial.

were neither published nor reported in a trial registry. The results of investigator-sponsored trials were more likely to be unavailable than industry-sponsored trials (24%; 35/147 vs. 4%; 7/179). Among unpublished trials, 40% (28/70) made their results available through a clinical trial registry (industry-sponsored 79% [26/33] vs. 5% [2/37] investigator-sponsored; **Table 2**).

Approximately 1 in 3 RCTs were prematurely discontinued (30%; 98/326; **Tables 2** and S2). From the discontinued RCTs, 67% (66/98) were available as a peer-reviewed publication (**S3 Table**). Compared to discontinued trials, completed RCTs were more likely to be published (OR 7.08; 95% CI: 3.35 to 15.52; $p < 0.001$) and to make their results available through a journal or trial registry (OR 6.06; 95% CI: 1.94 to 22.24; $p < 0.001$; **S3 Table**).

The main reason for RCT discontinuation was poor recruitment (37%; 36/98) followed by stopping for futility (16%; 16/98; **Table 3**). The proportion of prematurely discontinued RCTs did not change compared to protocols approved a decade earlier (28% of protocols approved in 2000 to 2003 [14] versus 30% of those approved in 2012). Data comparing results from this study with the cohort from 2000 to 2003 [14] are presented in **S4 Table**. Switzerland had more unregistered trials (7.9%; 13/165), compared to the other countries (UK 4.5%, 4/89; Germany 2.7%, 1/37; Canada 2.9%, 1/35); otherwise, results were comparable in terms of discontinuation and nonpublication among the 4 included countries (**S2 Table**). Multivariable analyses indicated that RCTs with more complete protocol reporting according to SPIRIT guidelines [19,20] were less likely to be unpublished (OR 0.71 [in increments of 10% proportion of adherence]; 95% CI: 0.55 to 0.92; $p = 0.009$; **Table 4**). None of the assessed characteristics were found to be associated with discontinuation due to poor recruitment or discontinuation due to preventable reasons. A separate analysis excluding the sample from the UK found no association of CTU/CRO support with trial discontinuation or publication (see **S5 Table**).

## Discussion

Our study found that most RCTs with ethics approval in 2012 (94%) were registered; however, 1 in 10 were registered retrospectively. Further, when restricted to unpublished RCTs, only 4 out of 5 trials were registered. RCT protocols with higher reporting quality, as indicated by greater adherence to SPIRIT guidelines [19,20], were more likely to be published in a peer-

**Table 1. Baseline characteristics of included RCTs.**

| | Industry-sponsored RCTs (*n* = 179) | Investigator-sponsored RCTs (*n* = 147) | All RCTs (*n* = 326) |
|---|---|---|---|
| Planned sample size, median (IQR)[a] | 360 (144–800) | 150 (60–426) | 250 (100–600) |
| Proportion of adequately reported SPIRIT items in protocol, median (IQR) | 0.74 (0.67–0.79) | 0.63 (0.54–0.70) | 0.69 (0.61–0.77) |
| Single center vs. multicenter | | | |
| Single center | 6 (3.4%) | 54 (36.7%) | 60 (18.4%) |
| Multicenter | 173 (96.7%) | 93 (63.3%) | 266 (81.6%) |
| Study design | | | |
| Parallel | 171 (95.5%) | 125 (85.0%) | 296 (90.8%) |
| Crossover | 4 (2.2%) | 9 (6.1%) | 13 (4.0%) |
| Factorial | 3 (1.7%) | 7 (4.8%) | 10 (3.1%) |
| Cluster | 0 (0%) | 4 (2.7%) | 4 (1.2%) |
| Other[b] | 1 (0.6%) | 2 (1.4%) | 3 (0.9%) |
| Placebo controlled | 94 (52.5%) | 37 (25.2%) | 131 (40.2%) |
| Recruitment projection reported in protocol | 43 (24.0%) | 56 (38.1%) | 99 (30.4%) |
| Research ethics committee approval | | | |
| Switzerland | 87 (48.6%) | 78 (53.1%) | 165 (50.6%) |
| United Kingdom | 45 (25.1%) | 44 (29.9%) | 89 (27.3%) |
| Germany | 26 (14.5%) | 11 (7.5%) | 37 (11.4%) |
| Canada | 21 (11.7%) | 14 (9.5%) | 35 (10.7%) |
| Intervention | | | |
| Drug | 152 (84.9%) | 55 (37.4%) | 207 (63.5%) |
| Medical devices | 20 (11.2%) | 33 (22.5%) | 53 (16.3%) |
| Surgical | 2 (1.1%) | 18 (12.2%) | 20 (6.1%) |
| Behavioral | 0 (0.0%) | 19 (12.9%) | 19 (5.8%) |
| Other[c] | 5 (2.8%) | 22 (15.0%) | 27 (8.3%) |
| Medical field | | | |
| Oncology | 44 (24.6%) | 16 (10.9%) | 60 (8.4%) |
| Surgical | 12 (6.7%) | 25 (17.0%) | 37 (11.4%) |
| Cardiovascular | 19 (10.6%) | 11 (7.5%) | 30 (9.2%) |
| Neurology | 17 (9.5%) | 8 (5.4%) | 25 (7.7%) |
| Other[d] | 87 (48.6%) | 87 (59.2%) | 174 (53.4%) |

[a]Missing data for planned sample size for 4 trial protocols was inserted from other sources (i.e., peer-reviewed publication; *n* = 3; trial registry; *n* = 1).

[b]Split body (*n* = 2), parallel group with 2 consecutive randomizations (*n* = 1).

[c]Dietary supplement, radiation, and rehabilitation.

[d]Anesthetics, dermatology, endocrinology, gastro/intestinal, gynecology, hematology, infectious diseases, intensive care, nephrology, orthopedics, pediatrics, psychiatry, respiratory, rheumatology, and ophthalmology.

IQR, interquartile range; RCT, randomized clinical trial; SPIRIT, Standard Protocol Items: Recommendations for Interventional Trials [19,20].

reviewed journal. Only 1 in 6 investigator-sponsored trials made results available in a trial registry. Approximately 1 in 3 RCTs were discontinued before the original planned sample size was reached, and 1 in 5 trials remained unpublished at 10 years follow-up. The results of discontinued RCTs were less likely to be published compared to completed trials. Reporting quality of trial protocols was not associated with premature discontinuation. In comparison to investigator-sponsored trials, industry-sponsored RCTs tend to perform better in prospectively registering trials, avoiding discontinuation due to poor recruitment, and making results available in trial registries.

Table 2. Registration, completion, and publication status of RCTs approved by research ethics committees in 2012.

| | Industry-sponsored RCTs (n = 179) N (%, 95% CI) | Investigator-sponsored RCTs (n = 147) N (%, 95% CI) | All RCTs (n = 326) N (%, 95% CI) |
|---|---|---|---|
| **Registration status** | | | |
| Registered | 175 (97.8%, 94.4%–99.4%) | 132 (89.8%, 83.7%–94.2%) | 307 (94.2%, 91.0%–96.5%) |
| Prospectively registered | 164 (91.6%, 86.6%–95.2%) | 110 (74.8%, 67.0%–81.6%) | 274 (84.0%, 79.6%–87.9%) |
| Retrospectively registered | 10 (5.6%, 2.7%–10.0%) | 22 (15.0%, 9.6%–21.8%) | 33 (10.1%, 7.1%–13.9%) |
| Not registered | 4 (2.2%, 0.6%–5.6%) | 15 (10.2%, 5.8%–16.3%) | 19 (5.9%, 3.5%–9.0%) |
| **Completion status** | | | |
| Completed | 119 (66.5%, 59.1%–73.3%) | 84 (57.1%, 48.7%–65.3%) | 203 (62.3%, 56.8%–67.6%) |
| Discontinued | 57 (31.8%, 25.1%–39.2%) | 41 (27.9%, 20.8%–35.9%) | 98 (30.1%, 25.1%–35.4%) |
| Unclear | 3 (1.7%, 0.3%–4.8%) | 22 (15.0%, 9.6%–21.8%) | 25 (7.7%, 5.0%–11.1%) |
| **Results availability** | | | |
| Peer-reviewed publication | 146 (81.6%, 75.1%–87.0%) | 110 (74.8%, 67.0%–81.6%) | 256 (78.5%, 73.7%–82.8%) |
| In clinical trial registry | 150 (83.8%, 77.6%–88.9%) | 23 (15.7%, 10.2%–22.5%) | 173 (53.1%, 47.5%–58.6%) |
| As peer-reviewed publication and in clinical trial register | 124 (69.3%, 62.0%–75.9%) | 21 (14.3%, 9.1%–21.0%) | 145 (44.5%, 39.0%–50.1%) |
| Results not available (neither as publication nor in clinical trial register) | 7 (3.9%, 1.6%–7.9%) | 35 (23.8%, 17.2%–31.5%) | 42 (12.9%, 9.4%–17.0%) |
| **Neither registered nor published** | 3 (1.7%, 0.3%–4.8%) | 12 (8.2%, 4.3%–13.8%) | 15 (4.6%, 2.6%–7.5%) |
| **Not published in journal but registered**[a] | 30/33 (90.9%, 75.7%–98.1%) | 25/37 (67.6%, 50.2%–82.0%) | 55/70 (78.6%, 67.1%–87.5%) |
| **Not published in journal but results available in registry**[a] | 26/33 (78.8%, 61.1%–91.0%) | 2/37 (5.4%, 0.7%–18.2%) | 28/70 (40.0%, 28.5%–52.4%) |

[a]Only a subsample of 70 unpublished trials considered.

CI, confidence interval; RCT, randomized clinical trial.

Table 3. Reasons for trial discontinuation and proportion of results available.

| Reasons for discontinuation | All Discontinued RCTs (n = 98) | Industry-sponsored discontinued RCTs (n = 57) | Investigator-sponsored discontinued RCTs (n = 41) | Results available as a peer-reviewed publication | Results available in clinical trial register | Results in peer-reviewed publication and clinical trial register | Results not available |
|---|---|---|---|---|---|---|---|
| Poor recruitment[a] | 36 (37%) | 16 (28%) | 20 (49%) | 21 (58%) | 16 (44%) | 11 (31%) | 10 (28%) |
| Futility | 16 (16%) | 15 (26%) | 1 (2%) | 11 (69%) | 13 (81%) | 9 (56%) | 1 (6%) |
| Harm | 6 (6%) | 5 (9%) | 1 (2%) | 5 (83%) | 6 (100%) | 5 (83%) | 0 (0%) |
| Organizational/ strategic reasons | 6 (6%) | 6 (11%) | 0 (0%) | 3 (50%) | 4 (67%) | 1 (17%) | 0 (0%) |
| Benefit | 3 (3%) | 2 (4%) | 1 (2%) | 3 (100%) | 2 (67%) | 2 (67%) | 0 (0%) |
| External evidence | 3 (3%) | 0 (0%) | 3 (7%) | 3 (100%) | 1 (33%) | 1 (33%) | 0 (0%) |
| Limited resources | 1 (1%) | 0 (0%) | 1 (2%) | 1 (100%) | 0 (0%) | 0 (0%) | 0 (0%) |
| Unclear | 27 (28%) | 13 (23%) | 14 (34%) | 19 (70%) | 12 (44%) | 6 (22%) | 2 (7%) |
| Discontinued due to a preventable reason[b] | 70 (71%) | 35 (61%) | 35 (85%) | 44 (63%) | 32 (46%) | 18 (26%) | 12 (17%) |

[a]Two studies that stated slow recruitment as reason for discontinuation mentioned in addition another reason (i.e., organizational/strategic reasons n = 1; external evidence n = 1).

[b]Counting the following reasons as not preventable: futility, harm, benefit, external evidence. Counting the following as preventable: poor recruitment, organizational/strategic reasons, limited resources, and unclear reasons (assuming that discontinuation due to unclear reasons was mainly due to non-data-driven reasons [18]).

RCT, randomized clinical trial.

**Table 4. Factors associated with (a) publishing main results in a peer-reviewed journal; (b) discontinuation of trials due to poor recruitment; and (c) discontinuation of trials due to preventable reasons.**

| Characteristics | | | Univariable | | | Multivariable | | |
|---|---|---|---|---|---|---|---|---|
| | | | OR | 95% CI | P value | OR | 95% CI | P value |
| **Nonpublication in a peer-reviewed journal** | **RCT not published in a peer-reviewed journal (n = 70)** | **RCTs published in a peer-reviewed journal (n = 256)** | | | | | | |
| Proportion of adequate SPIRIT reporting, median (IQR)[a] | 0.66 (0.53, 0.73) | 0.70 (0.62, 0.78) | 0.69 | 0.57–0.84 | <0.001 | 0.71 | 0.55–0.92 | 0.009 |
| Planned target sample size, median (IQR)[b] | 146 (60, 288) | 315 (109, 719) | 0.99 | 0.97–1.01 | 0.215 | 0.99 | 0.98–1.01 | 0.377 |
| Placebo controlled (vs. not placebo controlled) | 30/70 (42.9%) | 101/256 (39.5%) | 1.15 | 0.67–1.97 | 0.607 | 1.48 | 0.82–2.66 | 0.193 |
| Single center (vs. multicenter) | 20/70 (28.6%) | 40/256 (15.6%) | 2.26 | 1.16–4.01 | 0.015 | 1.35 | 0.64–2.86 | 0.434 |
| Reported recruitment projection | 15/70 (21.4%) | 84/256 (32.8%) | 0.56 | 0.30–1.05 | 0.069 | 0.75 | 0.38–1.49 | 0.409 |
| Industry sponsorship | 33/70 (47.1%) | 146/256 (57.0%) | 0.67 | 0.40–1.14 | 0.142 | 1.03 | 0.51–2.06 | 0.937 |
| **Discontinued due to poor recruitment** | **RCTs discontinued due to poor recruitment (n = 36)** | **RCTs not discontinued due to poor recruitment (n = 265)[c]** | | | | | | |
| Proportion of adequate SPIRIT reporting, median (IQR)[a] | 0.66 (0.60, 0.75) | 0.70 (0.63, 0.78) | 0.85 | 0.64–1.13 | 0.261 | 0.98 | 0.69–1.40 | 0.905 |
| Planned target sample size, median (IQR)[b] | 135 (79, 413) | 300 (108, 720) | 0.94 | 0.88–1.02 | 0.133 | 0.95 | 0.89–1.02 | 0.159 |
| Placebo controlled (vs. not placebo controlled) | 15/36 (41.7%) | 109/265 (41.1%) | 1.02 | 0.50–2.07 | 0.951 | 1.32 | 0.62–2.81 | 0.475 |
| Single center (vs. multicenter) | 8/36 (22.2%) | 39/265 (14.7%) | 1.66 | 0.70–3.90 | 0.248 | 0.93 | 0.34–2.50 | 0.883 |
| Reported recruitment projection | 11/36 (30.6%) | 78/265 (29.4%) | 1.05 | 0.49–2.25 | 0.890 | 1.08 | 0.43–2.52 | 0.862 |
| Industry sponsorship | 16/36 (44.4%) | 160/265 (60.4%) | 0.53 | 0.26–1.05 | 0.072 | 0.54 | 0.22–1.30 | 0.170 |
| **Discontinued due to preventable reasons** | **RCTs discontinued due to preventable reason (n = 70)[d]** | **RCTs not discontinued due to preventable reason (n = 231)[c,d]** | | | | | | |
| Proportion of adequate SPIRIT reporting, median (IQR)[a] | 0.68 (0.61, 0.73) | 0.70 (0.62, 0.78) | 0.82 | 0.66–1.02 | 0.080 | 0.94 | 0.72–1.24 | 0.668 |
| Planned target sample size, median (IQR)[b] | 169 (90, 500) | 315 (110, 718) | 1.00 | 0.99–1.01 | 0.549 | 0.99 | 0.99–1.01 | 0.747 |
| Placebo controlled (vs. not placebo controlled) | 28/70 (40.0%) | 96/231 (41.6%) | 0.94 | 0.54–1.62 | 0.816 | 1.01 | 0.59–1.89 | 0.859 |
| Single center (vs. multicenter) | 14/70 (20.0%) | 33/231 (14.3%) | 1.50 | 0.75–3.00 | 0.251 | 1.02 | 0.46–2.27 | 0.960 |
| Reported recruitment projection | 15/70 (21.4%) | 74/231 (32.0%) | 0.58 | 0.31–1.09 | 0.091 | 0.56 | 0.28–1.12 | 0.101 |
| Industry sponsorship | 35/70 (50.0%) | 141/231 (61.0%) | 0.64 | 0.37–1.09 | 0.102 | 0.63 | 0.32–1.24 | 0.183 |

[a]In increments of 10%.

[b]In increments of 100.

[c]Studies with unclear discontinuation status excluded.

[d]Counting the following reasons as not preventable: futility, harm, benefit, external evidence. Counting the following as preventable: poor recruitment, organizational/strategic reasons, limited resources, and unclear reasons (assuming that discontinuation due to unclear reasons was mainly due to non-data-driven reasons [18]).

CI, confidence interval; IQR, interquartile range; OR, odds ratio; RCT, randomized clinical trials; SPIRIT, Standard Protocol Items: Recommendations for Interventional Trials [19,20].

## Comparison with other studies

A systematic review and meta-analysis published in 2018 found that in different medical specialties, 2% to 79% of RCTs were not registered [21]. The study authors highlighted that the proportion of registered trials increased over time [21]. However, they only considered published RCTs, whereas we had access to the trial protocols and were also able to explore if unpublished results were made available in a registry. Overall, 6% of the RCTs from our sample were not registered. When separately assessing published and nonpublished RCTs, these proportions were 2% and 21%, respectively, indicating that nonregistration is more common among nonpublished RCTs. The proportion of prospectively registered RCTs was 84%. Other studies show a wide range of lower proportions (24% to 72%) [9,22], depending on medical specialties, time frame assessed, and journals considered for selecting included RCTs. We can only speculate why we found a higher rate of prospectively registered RCTs. Reasons might be that there was a general improvement over the last years and that the included countries from which we selected RECs might be more stringent in enforcing registration prior to patient recruitment.

Compared to a decade earlier [14], the proportion of prematurely discontinued RCTs did not change. Further, discontinued RCTs were more likely to remain unpublished in both our prior [14] and current study, and the most common reason for discontinuation remained poor recruitment. Publication rates between our cohorts showed improvement, with 59% of approved trials appearing in a peer-reviewed journal in our prior study [14] and 79% in our current analysis. When assessing the availability of results either as a publication or in clinical trial registry, 87% of study results were available (not assessed for RCTs approved in 2000 to 2003 [14]). In addition, both studies found that industry-sponsored trials published their results more frequently and were associated with lower rates of discontinued trials [14]. A 2018 systematic review concluded that industry-sponsored trial publications were more comprehensively reported than investigator-sponsored [23]. In line with these results, we found that the reporting quality in RCT study protocols approved in 2012 was better for industry-sponsored trials compared to investigator-sponsored RCTs. A systematic review by Schmucker and colleagues [24] revealed that 2 previous studies also assessed the publication rate of ethically approved RCT protocols. Both found low publication rates for protocols that were approved between 1988 and 1998 (i.e., 52%; 233/451; approved 1988 to 1998 in Switzerland [25]; 37%; 102/274 approved 1994 to 1995 in Denmark [26]). These findings are in line with our repeated metaresearch analysis, indicating that the publication rate has improved over the last decades.

## Strengths and limitations

Strengths of our study include full access to the protocols of all trials approved by the collaborating RECs during the study period. We recruited reviewers with training in healthcare methodology to complete all data abstraction and considered only a limited number of variables in our regression models to reduce the chance of spurious associations. Further, we sampled trial protocols from the same RECs in both 2000 to 2003 and 2012 (with the exception of the added UK REC in 2012), which provides greater confidence in the shifts we observed regarding increased rates of RCT discontinuation and improved rates of publication over the past decade.

Our study has the following limitations: First, our sample size was modest, which may have limited the ability of our regression models to identify significant associations between protocol features and discontinuation or nonpublication. Second, of the 19 RCTs we classified as unregistered, the completion status of 11 was unclear. Thus, it is possible that some of those

RCTs were never started and therefore the proportion of 6% unregistered RCTs may be smaller. Third, the SPIRIT checklist was created as a reporting guideline and not as a measurement tool for reporting quality [27]. However, we carefully operationalized the SPIRIT checklist and conducted various sensitivity analyses before using reporting quality estimates for the present study [15,17]. Fourth, we used trial protocols approved by RECs in Switzerland, the UK, Germany, and Canada, and the generalizability of our findings to RCT protocols approved by other RECs in these or other countries is uncertain. Fifth, regulatory aspects might have changed in some countries since 2012 (e.g., registration has been mandatory by law since 2014 in Switzerland, which had the highest proportion of unregistered trials [28,29]), hence it is possible that registration rates are higher nowadays.

## Implications

Our study revealed encouraging results in terms of registration rates and making trial results available, but further efforts are still needed. Meerpohl and colleagues have developed 47 recommendations targeted at a variety of stakeholders [30]. Among others, they strongly recommend that legislators make trial registration mandatory, funding agencies request dissemination of all funded projects, and that RECs require trial registration before the recruitment of the first patient and request annual reports describing the dissemination of study results. Furthermore, publishing journals should remove barriers to publish negative or inconclusive results (e.g., from discontinued trials) and trial investigators should consequently make results available in trial registries [30]. Currently few investigator-sponsored RCTs make their results available in trial registries. The advantage of results posted in a trial registry over results reported in a published article may be the avoidance of spin [31]. Future research should address the current hurdles that exist among investigators to share study results in trial registries and how sharing results in registries could be promoted.

Other areas, such as discontinuation of RCTs due to preventable reasons and retrospective registration, need to be addressed too. Future research should assess if the rate of discontinued trials due to poor recruitment can be reduced with pilot or feasibility studies [18]. As stated by clinical trial registry representatives, trial registration and prospective trial registration should be enforced by publishing journals (including checking if trial registration exists) [32]. In case a trial was not prospectively registered, authors should at least explain in the published article why this was not done.

## Conclusions

In our sample of RCTs approved by RECs from 4 countries, almost all were registered; however, 1 in 10 trials was registered retrospectively, which could result in methods being altered by study findings (e.g., changing the primary outcome [33]). Furthermore, 1 in 5 unpublished trials were not registered, and only 1 in 6 investigator-sponsored trials made results available in a trial registry. Higher reporting quality of trial protocols was positively associated with peer-reviewed publication, but not with prevention of trial discontinuation, highlighting the importance of feasibility assessments before embarking on a definitive trial. Despite a decade of efforts, premature trial discontinuation and nonpublication of RCTs remain common and comprise important targets to reduce waste in research.

## Supporting information

**S1 PRISMA Checklist. Preferred Reporting Items for Systematic Reviews and Meta-Analyses (PRISMA) checklist.**
(DOCX)

**S1 Text. Information on all participating research ethics committees.**
(DOCX)

**S2 Text. Procedure to receive more information about the trial status from ethical committees or by contacting principal investigators by sending them a survey through ethical committees.**
(DOCX)

**S3 Text. Survey to receive more information from investigators about the fate of their trial.**
(DOCX)

**S4 Text. Search strategy to identify corresponding full text publications.**
(DOCX)

**S5 Text. Code for analysis in Stata.**
(DOCX)

**S1 Table. Baseline characteristics of included randomized controlled trials stratified by country of ethical approval.**
(DOCX)

**S2 Table. Registration, completion, and publication status of randomized controlled trials approved in 2012 stratified by country of ethical approval.**
(DOCX)

**S3 Table. Association between completion of a randomized controlled trial and making the results available.**
(DOCX)

**S4 Table. Nonpublication and discontinuation in protocols approved by ethical committees in 2012 compared to protocols approved between 2000 and 2003.**
(DOCX)

**S5 Table. Factors associated with (a) publishing main results in a peer-reviewed journal; (b) discontinuation of trials due to poor recruitment; and (c) discontinuation of trials due to preventable reasons. UK samples excluded because variable "CTU/CRO support" was not assessed.**
(DOCX)

## Acknowledgments

All participating ethics committees were project partners. We are grateful for the support from RECs from Germany (Freiburg), Switzerland (Basel, Bellinzona, Bern, Geneva, Lausanne, St. Gallen, Thurgau, Zurich), Canada (Hamilton), and the UK (National Health Service Health Research Authority; Bristol office of the UK National Research Ethics Service) and thank them for their cooperation and for granting us access to approved study protocols from 2012. We are grateful to Prof. Doug Altman (Centre for Statistics in Medicine, University of Oxford) who initially conceived the concept of this study and who sadly passed away before it was completed.

## Author Contributions

**Conceptualization:** Benjamin Speich, Dmitry Gryaznov, Jason W. Busse, Erik von Elm, Belinda von Niederhäusern, Sally Hopewell, Ayodele Odutayo, Matthias Briel.

**Data curation:** Benjamin Speich, Dmitry Gryaznov, Jason W. Busse, Viktoria L. Gloy, Szimonetta Lohner, Katharina Klatte, Ala Taji Heravi, Nilabh Ghosh, Hopin Lee, Anita Mansouri, Ioana R. Marian, Ramon Saccilotto, Edris Nury, Benjamin Kasenda, Elena Ojeda–Ruiz, Stefan Schandelmaier, Yuki Tomonaga, Alain Amstutz, Christiane Pauli–Magnus, Karin Bischoff, Katharina Wollmann, Laura Rehner, Joerg J. Meerpohl, Alain Nordmann, Jacqueline Wong, Ngai Chow, Patrick Jiho Hong, Kimberly Mc Cord – De Iaco, Sirintip Sricharoenchai, Arnav Agarwal, Matthias Schwenkglenks, Lars G. Hemkens, Erik von Elm, Bethan Copsey, Alexandra N. Griessbach, Christof Schönenberger, Dominik Mertz, Anette Blümle, Belinda von Niederhäusern, Sally Hopewell, Ayodele Odutayo, Matthias Briel.

**Formal analysis:** Benjamin Speich.

**Funding acquisition:** Matthias Briel.

**Methodology:** Benjamin Speich, Matthias Schwenkglenks, Erik von Elm, Belinda von Niederhäusern, Sally Hopewell, Ayodele Odutayo, Matthias Briel.

**Project administration:** Benjamin Speich, Dmitry Gryaznov, Viktoria L. Gloy, Anette Blümle, Ayodele Odutayo.

**Resources:** Matthias Briel.

**Software:** Ramon Saccilotto.

**Supervision:** Sally Hopewell, Matthias Briel.

**Validation:** Benjamin Speich.

**Visualization:** Matthias Briel.

**Writing – original draft:** Benjamin Speich, Jason W. Busse, Matthias Briel.

**Writing – review & editing:** Dmitry Gryaznov, Viktoria L. Gloy, Szimonetta Lohner, Katharina Klatte, Ala Taji Heravi, Nilabh Ghosh, Hopin Lee, Anita Mansouri, Ioana R. Marian, Ramon Saccilotto, Edris Nury, Benjamin Kasenda, Elena Ojeda–Ruiz, Stefan Schandelmaier, Yuki Tomonaga, Alain Amstutz, Christiane Pauli–Magnus, Karin Bischoff, Katharina Wollmann, Laura Rehner, Joerg J. Meerpohl, Alain Nordmann, Jacqueline Wong, Ngai Chow, Patrick Jiho Hong, Kimberly Mc Cord – De Iaco, Sirintip Sricharoenchai, Arnav Agarwal, Matthias Schwenkglenks, Lars G. Hemkens, Erik von Elm, Bethan Copsey, Alexandra N. Griessbach, Christof Schönenberger, Dominik Mertz, Anette Blümle, Belinda von Niederhäusern, Sally Hopewell, Ayodele Odutayo.

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
