## [Editor Report · Decision Letter 0]

13 Dec 2021

Dear Dr Speich, 

Thank you for submitting your manuscript entitled "Non-registration, discontinuation, and non-publication of randomized trials: A repeated meta-research analysis" for consideration by PLOS Medicine.

I am writing to let you know that we would like to send your submission out for external peer review.

Please re-submit your manuscript within two working days, i.e. by Dec 15 2021 11:59PM.

Kind regards,

Callam Davidson

Associate Editor

PLOS Medicine

---

## [Decision Letter · Decision Letter 1]

12 Jan 2022

Dear Dr. Speich,

Thank you very much for submitting your manuscript "Non-registration, discontinuation, and non-publication of randomized trials: A repeated meta-research analysis" (PMEDICINE-D-21-05025R1) for consideration at PLOS Medicine. 

Your paper was evaluated by an associate editor and discussed among all the editors here. It was also discussed with an academic editor with relevant expertise, and sent to independent reviewers, including a statistical reviewer. The reviews are appended at the bottom of this email and any accompanying reviewer attachments can be seen via the link below:

[LINK]

In light of these reviews, I am afraid that we will not be able to accept the manuscript for publication in the journal in its current form, but we would like to consider a revised version that addresses the reviewers' and editors' comments. Obviously we cannot make any decision about publication until we have seen the revised manuscript and your response, and we plan to seek re-review by one or more of the reviewers. 

We hope to receive your revised manuscript by Feb 02 2022 11:59PM. Please email us (plosmedicine@plos.org) if you have any questions or concerns.

We look forward to receiving your revised manuscript. 

Sincerely,

Callam Davidson, 

Associate Editor

PLOS Medicine

plosmedicine.org

Comments from the academic editor:

I would suggest that the authors (1) better describe the sample so as to help understand if it is generalizable and (2) dig into the 'research waste' idea a bit more, since a major limitation of the study is any details on the RCTs themselves to understand how many patients accrued, what the flaws in the study were, etc. If no patients accrued, then where is the 'waste'?

Please report your SR/MA according to the PRISMA guidelines provided at the EQUATOR site.

http://www.equator-network.org/reporting-guidelines/prisma/

Please provide the completed PRISMA checklist. I appreciate that the nature of the study means some items may be non-applicable. 

Please add the following statement, or similar, to the Methods: "This study is reported as per the Preferred Reporting Items for Systematic Reviews and Meta-Analyses (PRISMA) guideline (S1 Checklist)."

If possible, please update your search end date from March 2021 to the present time.

The Data Availability Statement (DAS) requires revision. Please also include an appropriate contact (web or email address) for inquiries (this cannot be a study author).

Abstract Methods and Findings:

* Please include the study design.

* Please include the key databases searched.

* Please include relevant p-values when quantifying your results.

Abstract Conclusions:

* Please address the study implications without overreaching what can be concluded from the data; the phrase "In this study, we observed ..." may be useful.

Please remove the Funding and Competing Interests sections from the main text, this information is captured as metadata from the submission form.

Please define the abbreviations in Figure 1.

Please include the relevant p values for the results presented on lines 262 and 267.

Line 296: Please do not present new data in the Discussion, relocate this to the Results.

Please expand your Discussion to include implications and next steps for research, clinical practice, and/or public policy (this should follow your Strengths and Limitations section and precede your Conclusions).

Please relocate the Patient and Public Involvement section to the Methods.

Reference #5: Please remove the COI information from this reference.

Supplementary materials: Please update your supplementary references to follow our reference guidelines (https://journals.plos.org/plosmedicine/s/submission-guidelines#loc-references) 

Comments from the reviewers:

Reviewer #1: I thank the editor to having the opportunity to review this manuscript.

The authors report the fate (with regards to completion, quality, and publication) of 327 randomized controlled trials (RCTs) approved by research ethics committees in 4 countries in 2012. The assessment was conducted in 2021. 

This work is of high interest since there are very few studies that have access to the protocols approved by RECs ─most similar analysis only have access to the published (on articles or posted on registries) results. The results reported are also of interest since the same authors conducted a similar analysis with RCTs approved in 2003-2004 allowing to compare the outcome of both cohorts. The manuscript is very easy to follow, well written and focused. I would stress the highly useful data reported regarding reasons for discontinuation, which is hardly found in other reports.

This manuscript should affect public health policy with regards to the low fulfilment of posting trial results of investigator-initiated trials on trial registries. RECs should be involved on this.

I have no major concerns on how the data was presented and analyzed, except for the following: In Acknowledgement, one can read the RECs involved. However, it is not clear how the UK NHSHRA provided the 89 RCTs included in this analysis. I would bet that there were many more than 89 RCTs approved in the UK in 2012; then, which were the criteria to choose those 89 RCTs? This is a very relevant information that readers should know and that should be included in Methods.

In addition, I would like to stress that mentioning that investigator-initiated trials usually fail on having trial results posted on the registries should be included in both the abstract and conclusion. This should be one of the main messages of this report.

Below I mentioned a few suggestions that, I believe, would be of interest to many readers.

Minor comments: 

Seems to me that this research includes RCTs on medicines only (Industry-sponsored may also include trials on devices, but I believe that the research was conducted on RCTs assessing medicines only). That the research was conducted on medicines RCTs must be stated in the title, and in other appropriate sections of the manuscript.

Abstract

The abstract should report the number (and percentage) of RCTs by trial phase (2, 3 or 4) and the number (and percentage) of RCTs that were not published but of which the results are posted on the registries, and on RCTs that were published and posted the results on the registries. (This latter is relevant, since many RCTs could be published in subscription-only journals, so access to the results is limited to a minority of interested parties). 

Reporting the clinical development phases (2, 3 and 4) of the RCTs assessed is relevant, since the clinical usefulness of phase 3 and 4 RCTs is much higher than of phase 2 trials.

Introduction

In addition to the ICJME requirement of trial registration, authors should mention that both the EU and North American regulations (Canada, USA) requested registration of trials on medicines. I do not know whether the Swiss regulation requested registration as of 2012, but is likely it did.

Methods

Authors should report the number of research ethics committees by country involved in this research. 

Lines 210-211.- Authors should inform who, among the authors, were the 'pairs of reviewers'. 

Among those RCTs that were registered in more than one registry, how many posted the results in all registries?

Analysis

(General comment: Since authors had complete data for all RCTs, I do believe it is inappropriate to present confidence intervals as if these were point estimates randomly sampled from a larger population. Do authors agree with this?)

Line 221.- Are the pre-specified independent variables in their models based on the previous analysis by the same authors or by published reports of other investigators? 

Lines 225-228.- What was the reason supporting the inclusion of a post-hoc analysis? That this analysis was done excluding UK RCTs (as reported in line 264) should be included here. I wonder how relevant this post-hoc analysis is.

Line 230.- p>0.05 should be p<0.05 (?)

Results

Lines 240-241.- The number of RCTs approved by REC and clinical development phases (2, 3 or 4) should be included in Table 1. Although of less relevance, I would also suggest reporting whether the trials were conducted in hospitals and/or primary care centers ─ this info might be difficult to know in some trials.

Line 250.- 4 published trials were not registered. It would be of interest to know whether the journals, at the time of publication, were followers of the ICMJE requirements or not as per the info included in their websites.

Line 264.- 'A separate analysis...' should read as follows: 'A separate post-hoc analysis…'

Lines 267-268: I would suggest deleting 'compared to discontinued trials' that is mentioned at the beginning of the sentence (line 266)

Discussion

Line 279.- I do believe that posting the results on the registry is even more relevant than publishing them (see above). Hence, I strongly suggest mentioning that industry-sponsored trials tend to perform better in this regard as well. Maybe the emphasis should be on the bad performance of investigator-sponsored trials, rather than in the better performance of industry-sponsored trials. 

Lines 319-321.- I do not see the need to add this 4th limitation…since only 6% of RCTs were not registered (!). In addition, mentioning it could mislead readers as, I do believe, that in 2012 all 4 countries where the RCTs were conducted were requesting prospective registration of trials ─as well as the ICMJE requirements, mentioned in the Introduction. 

Line 326.- One or two of the many references that support the statement 'e.g., changing the primary outcome' should be included here.

Line 326.- The statement '1 in 5 unpublished trials were not registered' has not enough relevance to be mentioned here (conclusions), since 94% of trials were registered (!).

Table 2.- In the EU, to have a RCT authorized by the regulatory agency of the country involved, an EudraCT number must have been given before study start. Hence, I would expect that the 20 RCTs that were not registered and those 34 that were retrospectively registered should have been approved by non-EU RECs. Am I correct? I do believe that the country (ies) in which these 54 RCTs were approved should be included as a footnote. This is a relevant piece of information.

Table 3.- An additional column reporting 'results available in a peer reviewed publication + clinical trial register' would be of great interest. Although regulations request to post clinical trial results on the registries, it is well known ─and the results of this research show─ that many sponsors publish the results in articles but forget to post them on the registry.

Reviewer #2: See attachment

Michael Dewey

Reviewer #3: Thank you for the invitation to review this rigorous and interesting research article. In this manuscript, Speich et al. focus on an important issue - rates of completion and publication of clinical trials. As the authors note, trial discontinuation and non-publication contribute to substantial research waste. Over the past two decades, clinical trial registries have been implemented to improve transparency and journal policies related to trial registration have been strengthened. This evaluation builds upon a previous evaluation by similar authors, which found that one-quarter of 1017 RCTs approved between 2000-2003 were discontinued prematurely; 44% remained unpublished at a median of 12 years follow-up. In the current evaluation, which included 327 RCTs protocols approved in 2012 by research ethics committees in Switzerland, UK, Germany, and Canada, the authors found that 31% were discontinued prematurely; 24% remained unpunished at 9 years follow-up. Overall, the authors conclude the premature trial discontinuation has not improved in the past decade. Although non-publication of RCTs has improved, opportunities exist for improvement. Overall, this is an important manuscript, which builds upon a prior, seminal evaluation. The findings remain relevant to a broad clinical audience. 

Here are a number of suggestions for the authors and editors to consider:

First, as the authors note, the sample size is rather small. In particular, the sample is nearly a third of the size of the previous evaluation. In the published protocol, the authors note that they identified 450 protocols approved in 2012 and 402 protocols approved in 2016. Is there a reason that the 2016 protocols were not considered? In particular, the authors note that they "propose to compare RCT cohorts from 2012 and 2016 with RCTs approved 2000-2003…". If the authors have these data collected, a larger sample size would also strengthen the logistic regression-based analyses conducted, which may have been underpowered. 

Second, the authors may want to consider updating the discussion section to include a brief section with potential implications. Currently, the authors recap the main findings (paragraph 1), provide a comparison with other studies (paragraphs 2 and 3), and present strengths and limitations. However, opportunities exist to mention several implications and/or recommendations, given that there has been little improvement over the past two decades. 

Third, it appears as if new data are reported in the Discussion section that are not included in the Results section (e.g., Supplementary Table 5). In particular, the authors present a table comparing study-protocols approved in 2012 with study-protocols approved in 2000-2003 in the Discussion section (data from the previous publication). As the authors note in the protocol, one of their objectives was to "compare the RCT cohorts from 2012 and 2016 with RCTs approved 2000-2003". However, these comparisons are not described in the Method or reported in the Results of the manuscript. To improve clarity, the authors could consider updating the Results section to include information from the formal comparison between the two samples, especially if they hope to discuss these data in the Discussion section.

Lastly, some of the results from the reported multivariable analyses may be difficult to interpret. In particular, those focused on odds ratios based on increments of 10% proportion of adherence. Why were increments of 10% proportion adherence selected? ("guidelines were less likely to not published their results in a peer-reviewed journal (OR 0.73 (in increments of 10% proportion adherence); 95%CI: 0.57-0.93)). This feel rather arbitrary, especially given that the SPIRIT checklist has 33 items. 

Specific comments:

Page 5, Abstract:

Line 139: Overall, the abstract is very clear and concise. The odds ratio (0.73 for each 10% increment in the proportion of SPIRIT items met) was the only part that may need some refining (please see above).

Line 141: Perhaps the authors could rephrase "Non-publication of RCTs has improved" to increase clarity. For instance, the proportion of RCTs that remain unpublished has decreased. I think the "non-publication" and has "improved" makes it challenging to identify whether it is good or bad. 

Introduction: 

Line 150: The authors note that "A number of studies have explored the proportion of published RCTs that are registered (5-8); however, such investigations use published RCTs - and not RCT protocols - as the denominator and thus unable to assess publication bias." However, it is worth noting that previous studies have also looked at the proportion of registered RCTs that are published (e.g., https://www.bmj.com/content/344/bmj.d7292 ; https://jamanetwork.com/journals/jama/fullarticle/1840223 ; https://journals.plos.org/plosmedicine/article?id=10.1371/journal.pmed.1000144 ; etc. [disclosure: I am not a co-author on any of these manuscripts]. These studies assess publication bias, so it is worth mentioning these efforts and the corresponding results. 

Methods:

Line 166: It was great to see the authors transparently report that their analysis of discontinuation due to preventable reasons was not prespecified. What prompted the additional analysis? Perhaps, the authors could include a short paragraph (supplementary materials?) outlining any changes from the published protocol?

Line 191: I may have missed this, but how were the trials matched if they did not have the same title, abbreviation, etc? 

Results:

Line 227+: How did these characteristics compare to the previous sample? Do the authors have information about therapeutic areas? 

I may have missed this in the Results, but did the authors report the following in the text:

1. How many trials were prematurely discontinued were not published? 

2. The proportion of trials that remained unpublished at 9 years follow-up? This shows up in the Discussion, but I did not see it in the Results text. 

3. The proportion of prospectively registered RCTs (83%). This shows up in the discussion but I did not see it in the Results text. 

Discussion:

The authors not that "of the 20 RCTs we classified as unregistered, the status of 11 was unclear". Was this reported in the Results text? 

Figures/Tables:

Table 3: Perhaps the authors could stratify the rows in the table, since all the rows before the last one add up to 100%.

Reviewer #4: Many thanks for the opportunity to review this manuscript. Overall, this seems to be a very methodologically solid piece of work, meets the stated aims, and aligns with the published protocol. I also commend the authors on being clear about deviations throughout their manuscript. My comments are generally around minor clarifications and elaborations.

Abstract:

Page 6, lines 135-136: "We recorded whether RCTs were registered, discontinued (including reason for discontinuation), or published." Based on my understanding, this should say "and published" not "or published."

Introduction:

I'd like to see a little bit more discussion of the overall ASPIRE project and the role of this study within the larger project. No need for excruciating detail, as I know you clearly reference the overall project protocol, but the context is useful for readers to understand where this fits into the larger project and why the work was undertaken.

Methods:

Page 8 Line 189-193: Worth noting the ICTRP is not actually a registry but a meta-registry of entries from other registries (including the other ones you looked at). Also, I assume you actually used the EUCTR and not the back-end EudraCT system (which do not overlap perfectly as some things on EudraCT are not made public on the EUCTR).

Why did you include the additional post-hoc independent variable? Seems like some reasoning or justification should be provided.

Did you check for/account for duplicate registration?

Can the authors please note their data sharing plans in the manuscript? While some information from the RECs may not be able to be shared, it seems to me that the overall dataset, or at the very least any analytic code, should be able to be shared. I strongly encourage the authors to do so or explain why they are not sharing their code and data.

Results:

One cut of the results I think is clearly missing is how many results were unique to a registry, especially among discontinued trials. Since publication in a peer reviewed journal might at times be difficult for a prematurely ended study, this is where registries really have the ability to shine as a dissemination mechanism. It seems you have the ability to offer some data here and I think it would be valuable to present.

Since only 30.3% of your sample actually reported a recruitment projection in the protocol (per Table 1), how did you then use this as an independent variable in your regressions? Seems like you would have had a lot of missing data to contend with and don't describe this anywhere. Perhaps I'm misunderstanding?

Discussion:

The context of this paper in relation to existing work is very sparse. You cite a bit about registration and then compare your findings to your previous findings. I would like to see a bit more depth in terms of delving into the existing literature related to your findings, for instance on reporting of industry vs. non-industry findings. In addition, I don't see a citation of Schmucker et al. 2014 anywhere (https://pubmed.ncbi.nlm.nih.gov/25536072/) which seems a very important citation as it is, to my knowledge, the most recent systematic review that includes an analysis of studies of eventual publication of trials approved by RECs.

I'd also like to see some more engagement with the implications of these findings in the Discussion. The whole section feels a bit empty. I'm left, as a reader, with little indication as to what these findings mean for the larger clinical trial landscape and what changes the authors might like to see or recommend or reflect on what has been done. The most we get is a single sentence suggesting more feasibility studies.

Another potential limitation is your use of the SPIRIT checklist in a way that assumes equivalence of the points. Additionally, there have been objections raised to using checklists as indicators of quality as the authors have done so here (https://onlinelibrary.wiley.com/doi/full/10.1002/hsr2.165).

Minor Comments:

Page 8 Line 181: "Pairs of reviewers determined, independently and in duplicate" redundant to say pairs worked in duplicate.

Just to make the authors aware there are still some track changes in the supplement.

[LINK]

---

## [Decision Letter · Decision Letter 2]

24 Mar 2022

Dear Dr. Speich,

Thank you very much for re-submitting your manuscript "Non-registration, discontinuation, and non-publication of randomized trials: A repeated meta-research analysis" (PMEDICINE-D-21-05025R2) for review by PLOS Medicine.

I have discussed the paper with my colleagues and the academic editor and it was also seen again by three reviewers. I am pleased to say that provided the remaining editorial and production issues are dealt with we are planning to accept the paper for publication in the journal.

[LINK]

We look forward to receiving the revised manuscript by Mar 31 2022 11:59PM.   

Sincerely,

Callam Davidson, 

Associate Editor 

PLOS Medicine

plosmedicine.org

Requests from Editors:

Data Availability: Thank you for providing more details in your statement. I think it is important (if possible) to have the full details of the participating ethics committees in the manuscript – could these be provided in the Supplementary Materials? This item could be cited at around line 214. 

Line 137: Please begin this sentence ‘Study limitations include’ or similar. 

Please reorder you Appendices items such that they appear in sequential order throughout the text (Appendices 2 and 3 currently are cited before Appendix 1).

Please cite Appendix 4 (code) in the Methods.

Appendix Table 4: I can’t see the relevant flag for footnote c?

Please relocate the ‘Registration and Protocol’ section to the Methods. The ‘Availability of Data’ and ‘Authors’ Contributions’ can be removed as these are captured as metadata via the Submission Form. 

Comments from Reviewers:

Reviewer #1: I think the authors have appropriately addressed my suggestions or have provided plausible/correct answers

Reviewer #2: The authors have addressed all my points especially in the toning down of the treatment of discontinued trials.

Michael Dewey

Reviewer #4: Many thanks for the opportunity to review a revised version of this manuscript. The authors have adequately addressed all of my points. I present some very minor points related to clarifications or copy-edits below but shouldn't need to see a revised version of the entire manuscript as these should mostly be easily addressed.

Line 180-181: Those requirements also often included requirements not just to register but to report results to the registry.

Line 188: "For example, if a RCT is discontinued due to slow participant recruitment before the planned sample size is reached, the data typically do not allow to answer the primary research question." - I think the last part of the sentence might be missing a word? "…the data typically do not allow to answer…" doesn't make sense.

Lines 189-190: "Hence, it is crucial that all RCT results (also from discontinued trials) are made available so that evidence is not lost and no resource waste minimized." - I think the word "no" in "no resource waste minimized" should be deleted?

Line 232: "...European Union Clinical Trial Registry (EudraCT)," change acronym to EUCTR.

Lines 232-233: "(4) the International Standard Randomised Controlled Trial Number (ISRCTN) registry." - While that is what ISRCTN used to stand for, technically it doesn't stand for anything anymore. The name of the registry is just the ISRCTN. (See: https://www.isrctn.com/page/about)

Line 313-315: "Compared to discontinued trials, completed RCTs were more likely to be published (OR 7.08; 95%CI: 3.35-15.52; p<0.001) and making their results available (OR 6.06; 95%CI: 1.94-22.24; p<0.001 Appendix Table 3)." - Should read "and make their results available." 

Lines 323-326: "Multivariable analyses indicated that RCTs with more complete protocol reporting according to SPIRIT [16, 17] guidelines were less likely to not publish their results in a peer-reviewed journal (OR 0.71 [in increments of 10% proportion of adherence]; 95%CI: 0.55-0.92; p=0.009; 326 Table 4)." - This is an awkward sentence construction with the double negative ("...less likely to not publish…"). Wouldn't it be accurate to say "RCTs with more complete protocol reporting according to SPIRIT were more likely to publish their results"? 

Lines 345-346: When discussion the context of unregistered RCTs, this review might be worth considering/including (perhaps in addition whatever individual studies you consider most notable): https://bmcmedicine.biomedcentral.com/articles/10.1186/s12916-018-1168-6

Lines 365-366: "Furthermore, it was shown several times that industry-sponsored trial publications are more comprehensively reported [19], what we also found is true for trial protocols approved in 2012." - This is an awkward sentence construction.

Line 379: Typo: "RTC" should say "RCT"

One other implication I'd like to potentially see discussed:

Lines 348-350: Overall, 6% of the RCTs from our sample were not registered. When separately assessing published and non-published RCTs, these proportions were 2% and 21%, respectively, indicating that non-registration is more common amongst non-published RCTs.

This seems to me to be a very important finding. If trials that don't register, also don't report, they are essentially invisible to everyone. This is the absolute worst outcome for everyone involved. That is a massive discrepancy in terms of percent between the two is probably worth highlighting.

[LINK]

---

## [Editor Report · Decision Letter 3]

1 Apr 2022

Dear Dr Speich, 

On behalf of my colleagues and the Academic Editor, Dr Aaron S Kesselheim, I am pleased to inform you that we have agreed to publish your manuscript "Non-registration, discontinuation, and non-publication of randomized trials: A repeated meta-research analysis" (PMEDICINE-D-21-05025R3) in PLOS Medicine.

When making the formatting changes please also make the following update:

* Include a reference for the citation 'Gryaznov et al., BMJ Open, in press' - per our submission guidelines, accepted, unpublished articles can be referenced in the bibliography in the same style as published articles, but substituting “Forthcoming” for page numbers or DOI.

PRESS

Sincerely, 

Callam Davidson 

Associate Editor 

PLOS Medicine